# Intrapleural Fibrinolytic Interventions for Retained Hemothoraces in Rabbits

**DOI:** 10.3390/ijms25168778

**Published:** 2024-08-12

**Authors:** Christian J. De Vera, Jincy Jacob, Krishna Sarva, Sunil Christudas, Rebekah L. Emerine, Jon M. Florence, Oluwaseyi Akiode, Tanvi V. Gorthy, Torry A. Tucker, Karan P. Singh, Ali O. Azghani, Andrey A. Komissarov, Galina Florova, Steven Idell

**Affiliations:** 1Department of Cellular and Molecular Biology, School of Medicine, The University of Texas Health Science Center at Tyler, 11937 US HWY 271, Tyler, TX 75708, USA; christianjordan.devera@uttyler.edu (C.J.D.V.); jincy.jacob@uttyler.edu (J.J.); krishna.sarva@uttyler.edu (K.S.); christudas.sunil@uttyler.edu (S.C.); remerine@patriots.uttyler.edu (R.L.E.); jon.florence@uttyler.edu (J.M.F.); oluwaseyi.akiode@uttyler.edu (O.A.); tanvigorthy013@gmail.com (T.V.G.); torry.tucker@uttyler.edu (T.A.T.); andrey.komissarov@uttyler.edu (A.A.K.); galina.florova@uttyler.edu (G.F.); 2Department of Epidemiology and Biostatistics, School of Medicine, The University of Texas Health Science Center at Tyler, 11937 US HWY 271, Tyler, TX 75708, USA; karan.singh@uttyler.edu; 3Department of Biology, The University of Texas at Tyler, 3900 University Boulevard, Tyler, TX 75799, USA; aazghani@uttyler.edu

**Keywords:** fibrinolysins, retained hemothorax, pleural disease, single-chain urokinase plasminogen activator (scuPA), single-chain tissue plasminogen activator (sctPA), plasminogen activator inhibitor-1 (PAI-1), intrapleural organization, pleural adhesions

## Abstract

Bleeding within the pleural space may result in persistent clot formation called retained hemothorax (RH). RH is prone to organization, which compromises effective drainage, leading to lung restriction and dyspnea. Intrapleural fibrinolytic therapy is used to clear the persistent organizing clot in lieu of surgery, but fibrinolysin selection, delivery strategies, and dosing have yet to be identified. We used a recently established rabbit model of RH to test whether intrapleural delivery of single-chain urokinase (scuPA) can most effectively clear RH. scuPA, or single-chain tissue plasminogen activator (sctPA), was delivered via thoracostomy tube on day 7 as either one or two doses 8 h apart. Pleural clot dissolution was assessed using transthoracic ultrasonography, chest computed tomography, two-dimensional and clot displacement measurements, and gross analysis. Two doses of scuPA (1 mg/kg) were more effective than a bolus dose of 2 mg/kg in resolving RH and facilitating drainage of pleural fluids (PF). Red blood cell counts in the PF of scuPA, or sctPA-treated rabbits were comparable, and no gross intrapleural hemorrhage was observed. Both fibrinolysins were equally effective in clearing clots and promoting pleural drainage. Biomarkers of inflammation and organization were likewise comparable in PF from both groups. The findings suggest that single-agent therapy may be effective in clearing RH; however, the clinical advantage of intrapleural scuPA remains to be established by future clinical trials.

## 1. Introduction

Intrapleural fibrinolytic therapy (IPFT), alternatively known as intrapleural enzymatic therapy, has been used since the late 1940s to clear pleural collections and enhance pleural drainage, initially in cases of intrapleural infections [1,2]. In clinical practice, retained hemothorax (RH) is generally cleared using either IPFT or surgical procedures to prevent subsequent fibrothorax, lung restriction, dyspnea, and infectious complications, including empyema. The use of IPFT has been adopted to avoid surgery, relieve pleural drainage, and prevent fibrotic complications [3,4,5,6,7]. IPFT is also useful for patients who are poor surgical candidates. While larger, randomized, controlled trials are still needed to conclusively document the place of IPFT in the treatment of RH; the consensus of recent literature currently supports the initial use of IPFT to clear RH [3,5,6].

Several issues relating to the use of IPFT remain poorly understood and represent gaps in the field [5,8,9]. The optimal choice of therapy, its dosage, and dosing schedule remain unclear. The same is true of the use of IPFT for the treatment of pleural infection with loculation and impaired pleural drainage [10,11]. In this study, we sought to test the ability of intrapleural administration of a single-chain urokinase (scuPA) to clear RH. We sought to identify an effective dosage range and test its safety. We also performed the first studies of which we are aware to assess schedules in RH. We compared these effects to the ability of single-chain tissue plasminogen activator (Activase, sctPA), which can be used off-label in RH. A range of assessments were used to assess clearance and clot resolution, including chest ultrasonography, two-dimensional clot measurement as well as clot displacement measurements, chest computed tomography (CT) scanning, and gross examination. Because sanguinous PF may promote local scarring, we also tested the effect of scuPA and sctPA to expedite pleural drainage through an indwelling thoracostomy tube. We used a recently established model of RH in the rabbit to conduct these studies [12]. The use of this model offers several advantages, as it is large enough to accommodate the insertion of an indwelling chest or thoracostomy tube and is amenable to serial chest imaging studies and gross examination of pleural organization, adhesion formation, pleural thickening, and collection of pleural tissues for histologic and immunohistology assessments. The rabbit is also amenable to intrapleural interventions with human proteins (scuPA or sctPA), which can be dosed repeatedly. The dosage of the scuPA and dosing schedule we used can be compared with those we previously used in rabbits to identify efficacy and safety in models of pleural empyema and organization [10,13]. By extrapolation, these studies can be presented to regulatory agencies and used to inform clinical trial dosing of scuPA, as recently reported [14]. We anticipate future clinical trial testing of scuPA in patients with RH.

## 2. Results

### 2.1. Experimental Design

The development of the RH model (Figure 1) was initiated by harvesting blood from donor rabbits. The homologous blood was mixed with citrate (1 mL of 5% citrate: 6 mL of blood) until RH induction. A chest tube was surgically placed in the right pleural space of each experimental rabbit, while CT was used to confirm placement and visualize the presence of iatrogenic pneumothorax. To induce a RH, the harvested citrated homologous blood was recalcified (CaCl_2_ 10 mM), supplemented with thrombin (0.3 U/mL), and immediately delivered into the pleural space via the placed chest tube. RH injury induction was performed in three delivery schedules (Figure 1, blood drops) to simulate continuous extravascular bleeding in the pleural space, delivering a total of 120 mL. A single chest tube was placed for the duration of each experiment, from the induction of RH through day 8 when the effects of the interventional agents were assessed. The chest tube was maintained via vest placement, and patency was maintained by performing a saline flush daily. The RH injury was monitored via chest ultrasonography.

After 7 d, the developed RH was treated with the fibrinolytic single-chain (sc) tPA (Activase) or scuPA (Figure 1) via the placed chest tube. The treatment dose and schedule include the following: phosphate-buffered saline (PBS, vehicle control) 2 mg/kg sctPA bolus or double delivery, 2 mg/kg scuPA bolus or double delivery, and 1 mg/kg scuPA double delivery only. A bolus dose was delivered at 0 h (Figure 1, Day 7). For the double dose schedule, a half dose was delivered at 0 h, with the second dose after 8 h. Aspiration via the previously placed chest tube was performed at 8 h and 24 h after intrapleural administration of scuPA or sctPA (at d7) to drain RH PFs, which otherwise did not drain via the closed chest tube. After 24 h, the efficacy of IPFT was assessed via the following outcomes: clot size reduction via ultrasonography and postmortem gross imaging, pleural drainage success, and lung volume expansion via CT (Figure 1). Histology was performed on lung tissues to evaluate pleural thickening and collagen deposition. A battery of assays was employed to look at the biochemical profile of RH PFs.

### 2.2. IPFT Successfully Reduced Clot Retention

The treatment of RH via IPFT considerably reduced clot retention. Representative sonographic and gross images confirmed the effective reduction in clot size with both a bolus and a double dose of fibrinolysins over 24 h (Figure 2A). Measurement of clot sizes (2D clot area, cm^2^) before (0 h) and after treatment (24 h) showed minimal change with PBS (−3.5 ± 6.7%) (Figure 2B,C). In contrast, a bolus dose of 2 mg/kg sctPA or scuPA achieved substantial clot size reductions of −57.8 ± 24.3% and −58.6 ± 16.0%, respectively (Figure 2B,C). Notably, a double dose of 2 mg/kg sctPA resulted in a −95.0 ± 12.37% reduction in clot size, while a double dose of 2 mg/kg scuPA, though slightly less effective, still produced a statistically significant reduction of 90.2 ± 11.6%, almost clearing the RH (Figure 2B,C). Since delivering a double dose of fibrinolytics appeared to be more effective, we performed another experiment with the same strategy while decreasing the dose of scuPA to 1 mg/kg. The sonographic findings showed that treatment with a half of effective dose of scuPA resulted in about −33.8 ± 18.1% clot size reduction (Figure 2B,C). To further validate our sonographic findings, the clots from treated subjects were harvested postmortem and measured via the volume displacement technique. The clots were not harvested at the earlier stages of the project to preserve adherence structure for the histological assessments, thus explaining the lack of data points for the bolus treatment group (Figure 2D). PBS treatment yielded a higher clot volume of 12.5 ± 2.7 mL, indicative of little to no changes in the pleural space. A bolus dose of 2 mg/kg sctPA or scuPA was less effective in reducing clot retention, having a clot volume of 9.5 ± 7.7 mL and 6.3 ± 4.7 mL, respectively. Lastly, treatment with a double dose schedule with 1 mg/kg sctPA or scuPA per injection (2 mg/kg total dose of plasminogen activator) resulted in a statistically significant lower clot volume of 0.08 ± 0.2 mL and 1.7 ± 2.1 mL, respectively, leading to nearly complete resolution of the RH in pleural space (Figure 2D). A further decrease in the dose of scuPA (0.5 mg/kg per dose) in the double dose schedule had a clot volume of 6.0 ± 2.4 mL, demonstrating a similar efficacy to a bolus dose of 2 mg/kg scuPA (Figure 2D).

### 2.3. IPFT Restored Pleural Drainage and Mitigate Lung Restriction

We assess the outcomes of IPFT—pleural drainage success, lung restriction mitigation, and lung re-expansion. The CT scans demonstrate a representative depiction of an injured rabbit that had successful outcomes with IPFT (Figure 3A). Before pleural drainage, we observed considerable lung restriction, as indicated by the increased opacification on both lungs (Figure 3A). After a successful drainage, opacity on both lungs were greatly reduced, representing the re-expansion of the lungs (Figure 3A). Pleural drainage success was defined as successful evacuation of PF via tube thoracostomy at 8 h and 24 h (Figure 3B). Treatment with PBS resulted in failed drainage with no or minimal (less than 1 mL) PFs evacuated out of the pleural space (Figure 3B). In contrast, treatment with either a bolus or double dose of 2 mg/kg sctPA or scuPA enabled pleural drainage, as indicated by the increase in the total evacuated PF; moreover, no comparable differences in total evacuated PFs were observed between these groups (Figure 3B). Notably, delivering 1.0 mg/kg scuPA as a double dose still successfully enabled pleural drainage; however, this group had a lower total evacuated PF volume (Figure 3B). Total lung volume at baseline, before drainage, and after drainage was monitored and quantified to assess the extent of lung restriction mitigation and subsequent lung re-expansion. The PBS treatment group did not receive a second CT due to the failed drainage (Figure 3C). PBS treatment demonstrated no statistical difference between lung volumes at baseline and before drainage, indicative of the lack of clot dissolution and formation of pleural effusion (Figure 3C). Comparably, the injured rabbits that received IPFT developed pleural effusion, resulting in a decrease in total lung volume (Figure 3C). Bolus and double doses of 2 mg/kg sctPA demonstrated similar efficacy in restoring lung volume (Figure 3C).

### 2.4. Pleural Thickening Was Unchanged in Rabbits Receiving IPFT

We next sought to ascertain the scope of pleural organization after intrapleural administration of scuPA and sctPA IPFT. The H&E and Trichrome staining of lung tissues of rabbits treated with IPFT (Figure 4A1–A8) were used in morphometry assessments. We measured visceral pleural thickening, which increased on day 7 after the induction of RH [12]. In contrast, a normal pleural lining would be a single cell layer, measuring about 50 µm, as previously shown [12]. While the clot burden of organizing RH clots was decreased by the delivery of scuPA and sctPA (Figure 2 and Figure 3), visceral pleural thickening was unchanged by the administration of either form of IPFT with fibrinolytic or vehicle control in rabbits with RH (Figure 4B).

### 2.5. PF Myeloid Cell Counts Were Comparable in RH Rabbits Treated with IPFT

Total red blood cell (RBC, Figure 5A) and white blood cell (WBC, Figure 5B) counts in the RH PFs were evaluated to assess whether IPFT led to increased extravascular bleeding and WBC infiltration. Additionally, WBC differential staining and analysis (Figure 5C–F) were conducted to elucidate the immune response to the RH treatment.

RBC and WBC counts in PFs of rabbits subjected to different doses of scuPA or sctPA via IPFT were similar, as shown in Figure 5A,B. RBC counts (Figure 5A) for the bolus and double doses of sctPA were marginally lower than those for the corresponding doses of scuPA. This observation did not correlate with intrapleural bleeding or other adverse effects, as RBCs were also present in the PFs of rabbits administered PBS (vehicle control). Only 3 rabbits in a group of 6 treated with PBS had an appreciative volume of pleural effusion, as displayed in Figure 5.

There were no significant differences in total WBC counts (Figure 5B) in PFs from rabbits with RH exposed to PBS or various dosing regimens of scuPA or sctPA. Neutrophil counts (Figure 5C) in the scuPA and sctPA treated groups were slightly elevated compared to the PBS-treated group, while lymphocyte counts (Figure 5D) were slightly reduced in both scuPA and sctPA groups compared to PBS. Eosinophil (Figure 5E), basophil (Figure 5F), and macrophage/monocyte (Figure 5G) counts were consistent across all groups.

### 2.6. Inflammatory Profile of RH PFs after IPFT Treatment

Pro-inflammatory (IL-6, IL-8, TNFα) and anti-inflammatory (TGF-β) cytokines in PFs were assessed via ELISA explore the role of these cytokines in regulating the expression of plasminogen activator inhibitor-1 (PAI-1), a major PA inhibitor, and to understand the changes in immune response, inflammation, and tissue repair upon RH treatment. Our findings indicated that IL-6 (Figure 6A), IL-8 (Figure 6B) levels were consistent across all groups. TNF-α levels were slightly elevated in groups that received double doses of both sctPA and scuPA compared to others (Figure 6C). TGF-β levels for the bolus 2 mg/kg dose of scuPA were significantly lower than those for PBS and the double dose of scuPA (1 mg/kg). Moreover, double doses of both sctPA and scuPA exhibited notably higher TGF-β expression than their respective bolus dose group (Figure 6D). 

### 2.7. Intrapleural Levels of Plasminogen Activators (tPA and uPA) and PAI-1 during IPFT of RH Model

We also performed ELISA analyses on tPA, uPA, and PAI-1 levels in RH PFs to evaluate the efficacy of our IPFT protocols. Our findings revealed that total PAI-1 levels remained consistent across the PBS control, bolus doses, and double doses of sctPA (2 mg/kg), as well as the bolus dose of scuPA (2 mg/kg). However, they were slightly elevated in the double doses of scuPA (1 and 2 mg/kg). Active PAI-1 levels were nearly zero in most rabbits treated with sctPA and scuPA, contrasting with slightly but not significantly elevated levels in the PBS-treated group. In the bolus treatments of sctPA and scuPA (2 mg/kg), active plasminogen levels were lower compared to their respective double doses. Furthermore, in the sctPA groups (both bolus and double dose), active PA levels at 8 h and 24 h post-treatment were comparable, whereas in the scuPA groups (2 mg/kg), active PA levels considerably decreased by 24 h compared to 8 h post-treatment. Total antigen levels of plasminogen activator were notably higher at 8 h compared to 24 h across all treatment groups. Additionally, scuPA demonstrated slightly higher total antigen levels of PA compared to sctPA.

## 3. Discussion

Rabbit models of pleural injury recapitulate disease conditions that are seen in human patients [12,16,17,18,19,20,21,22,23,24]. The similarity in the structure of fibrin [25] and the fibrinolytic system to those in humans make rabbit models suitable for fast testing of human plasminogen activators. Thus, translational rabbit models of pleural injury were considered for the design of clinical trials such as MIST2 [26] and LTI-01 [14], where sctPA and scuPA, respectively, were used to treat empyema. Thus, the results of the present study form a solid foundation for bench-to-bedside translation, supporting further preclinical studies of IPFT for RH, with a perspective of future clinical trials.

Our findings show that IPFT with scuPA or sctPA effectively accelerates clearance of intrapleural collections of organizing clots and improves lung restriction in rabbits with RH. Pleural drainage was also increased with the delivery of IPFT. The interventional agents were diluted in saline for intrapleural delivery [10]. There was no appreciable resolution of the RH clot burden in control rabbits exposed to the intrapleural saline (PBS) vehicle alone. The efficacy of scuPA and sctPA was increased in both bolus and two delivery platforms at a total dose of 2 mg/kg of either agent. Our data and clinical findings suggest that IPFT is effective and well-tolerated under the preclinical testing conditions we used. While the rabbit RH model only had pleural trauma attributable to chest tube placement, the literature described in the Introduction and this section indicates that IPFT is likewise generally well-tolerated in clinical practice, including post-operative settings. There are studies suggesting the possibility that larger doses of IPFT are required to effectively treat RH than empyema [4], but we found that similar doses of IPFT were effective in this study. Whether increased doses of IPFT are required in patients with RH will need to be evaluated in future clinical trial testing since the diversity of underlying conditions and bleeding could affect efficacy or safety. Our data suggest that the design of future clinical trials could involve a range of doses of scuPA comparable to those recently studied in empyema [14]. If sctPA is used, a new drug IND (investigational new drug approval) would be required for FDA approval, but dosing could be comparable.

We found that sctPA was as effective in clearing RH, improving lung restriction, and increasing pleural drainage as scuPA. Both agents were well tolerated by the rabbits, and no intrapleural bleeding complications were identified in the doses we used. We tested intrapleural scuPA and sctPA at a single bolus dose of 2 mg/kg or at two doses delivered 8 h apart on day 7. We tested scuPA treatment in the dose-de-escalation study (0.5 mg/kg delivered twice) and found that RH was partially cleared at this dose. Thus, our data clearly demonstrate that responses to IPFT are dose-dependent.

Intrapleural organization was reduced after administration of scuPA or sctPA IPFT, as PF drainage was consistently increased, and lung volume was increased after the efficient drainage. We speculate that the persistence of RH reflects the administered donor blood and local irritation of the thoracostomy tube.

The mechanism by which dual dosing of scuPA or sctPA was more effective in clot reduction than single bolus dosing was explored, but further investigation is needed to define the mechanism with precision. The levels of active PAI-1, the major inhibitor of plasminogen activators, were comparable in the RH PFs. We speculate that the bioavailability of scuPA and sctPA was likely increased in the rabbits that received dual dosing of the plasminogen activators. If so, increased bioavailability of the interventional agents could augment the reserve of intrapleural PA activity, increasing local fibrinolysis and more effectively clearing RH in the rabbits. The data suggest that the RH clot collection is amenable to early intervention and produces nearly complete resolution with this regimen; furthermore, these effects can be achieved by scuPA or sctPA without adjuncts. Whether repeat dosing of IPFT yields similar benefits in patients remains to be determined in future clinical trial testing.

The comparable PF red cell counts in controls and rabbits treated with interventional agents demonstrate that the incidence of intrapleural bleeding was not increased after administration via the thoracostomy tube (Figure 5A). Gross pleural hemorrhage was not seen after administration of scuPA or sctPA, nor was visceral or parietal bleeding found by histologic analyses (data not shown).

The selected cytokine and inflammatory mediator profiles of the PF after IPFT were comparable to those observed in samples collected prior to the administration of scuPA or sctPA. Similarly, these profiles were little changed after the administration of scuPA or sctPA at one or two doses, indicating that the inflammatory milieu of the PFs was not substantively altered by the administration of the fibrinolysins. The selected mediator composition of the PFs persisted; however, PF was increased as anticipated by intrapleural administration of the fibrinolysins. The levels of biomarkers measured in PFs at baseline agree well with those observed during model development [12]. Interestingly, baseline levels of TNF-α, IL-6, and IL-8 were in the ranges previously observed for rabbit models of chemically induced pleural injury [27] and advanced 7 d empyema [15]. On the other hand, levels of PAI-1 in RH were similar to those for the chemical model pleural injury model [27], but lower than PAI-1 in acute 3 d [27] and advanced 7 d [15] empyema models. In contrast, levels of TGF-β in RH were higher than those in the chemical model [27] and approached levels observed in the infectious model of pleural injury [15,27]. Levels of TGF-β were relatively lower in rabbits treated with 2 mg/kg single-bolus sctPA or scuPA. While the cause of the decrements remains unclear, lower levels of this growth factor could conceivably influence pleural organization, mesomesenchymal transition, or neovascularization although RH dissolution was observed in rabbits treated in this manner. Indeed, relatively high levels of active plasminogen activators (PA, 0.1–1.0 µg/mL; 1.6–20 nM) in combination with low levels of active PAI-1 (0–1.0 nM) were observed at 8 or 24 h after the treatment of the RH model with either a bolus or a double injection of fibrinolysins (Figure 7). Decrements in pleural fluid PA activity, including trends in lower active scuPA and total scuPA or sctPA, likely reflect the formation of complexes with active PAI-1 or clearance of these complexes. The relationship between these changes and interventional outcomes is unclear at present. These results are dramatically different from those observed with the rabbit model of *Streptococcus pneumoniae*-induced pleural injury [15,27], where PA activity was completely suppressed at 24 h after the treatment with sctPA or scuPA (bolus injection; up to 4 mg/kg) by increased expression of PAI-1. Therefore, in contrast to empyema, adjuncts such as DNase [26] or PAI-1 targeting [15,27,28,29] may not increase the efficacy of fibrinolysins in the treatment of RH lacking high intrapleural levels of PAI-1 and bacterial DNA. However, in cases of RH complicated by infection, which have been reported in clinical practice, treatment formulation and schedule may require addressing accumulated PAI-1 and DNA associated with the infection.

Pleural thickening was unchanged after the administration of scuPA and sctPA, while clot burden was decreased based upon two-dimensional ultrasonographic measurement and clot displacement analyses. This may reflect the persistent reaction of the pleural surfaces to the blood challenges and thoracostomy tube placement, which could have elicited a local inflammatory response. The clotted material in control rabbits was organized 7 days after induction of RH, as we previously observed adhesions and early fibrosis in the RH model [12]. With a reduction in the clot burden in the rabbits that received intrapleural scuPA or sctPA, less organization of RH was seen at gross inspection, as expected.

This study has inherent strengths and limitations. The rabbit model recapitulates the clot formation and intrapleural organization, lung restriction, and impaired pleural drainage that are seen in patients. Another strength is that an indwelling thoracostomy tube is present. However, the model is induced by the uniform administration of blood in three bolus injections of donor blood to induce RH formation in rabbits, while bleeding diatheses or slow but persistent bleeding may cause RH in patients [3,8]. The RH formed at 7 days is relatively nascent and so may be more amenable to local fibrinolysis despite the documented organization of the clot collection [12]. Bleeding risk could be increased in patients with chest trauma and influence the response to the administration of IPFT. This clinical bleeding propensity is not recapitulated in the model. In our view, these reservations are offset by advantages including persistent RH, the presence of the thoracostomy tube, the proximity of the rabbit fibrinolytic system to that of humans, and the ability to tolerate human protein interventions. Interestingly, intrapleural sctPA has been reported to induce hemothorax or thoracic bleeding [26,30]. In a recent safety trial, scuPA was well tolerated and did not induce hemothorax [14]. Whether this promising advantage is sustained in larger cohorts of patients receiving intrapleural scuPA requires future clinical trial testing.

This is the first study to document the ability of scuPA to clear RH and improve lung restriction and pleural drainage. The responses to the intrapleural administration of scuPA were comparable to that of sctPA in the model, information that adds to our understanding of how the relatively nascent RH coagulum can largely be cleared by 8 days after induction of RH in the model. These findings are consistent with the clinical literature, in which there have been several reports of the efficacy of initial administration of sctPA, two-chain urokinase, or other agents to clear RH [5,7,31,32,33,34,35,36,37,38]. These observations are also consistent with the prior finding that streptokinase was effective in clearing blood collections in ewes [36,37,38,39]. Whether adjuncts such as DNase or PAI-1 inhibitors can accelerate clearance of RH remains at issue and deserves further investigation. However, the addition of adjuncts may be unnecessary in initial clinical trials, based on our findings. Our findings suggest that single-agent therapy with scuPA or sctPA is very effective and well-tolerated in the model.

While our data support the use of IPFT in RH, there is still controversy as to whether early surgery is more effective than IPFT in the treatment of RH in clinical practice [40,41,42]. Early video-assisted thoracostomy (VATS) 3–6 days after identification of RH has been advocated to improve clinical outcomes and mitigate the potential for infectious complications such as empyema [7,41,43,44]. On the other hand, the literature offers an alternative perspective that IPFT can initially be attempted to clear RH, and surgery can be performed if IPFT is unsuccessful [7,45]. The current study does not address this debate. Rather, our findings support the contention that initial IPFT can be effective in clearing persistent RH and that a new agent, scuPA, can effectively clear RH with comparable efficacy to sctPA at the same doses in a rabbit RH model.

## 4. Materials and Methods

### 4.1. Animals

Pathogen-free New Zealand White Female rabbits (3.0 kg, 3 months) were purchased from Charles River (Wilmington, MA, USA). As previously described, we deployed conventional husbandry standards, provided food and water, and housed rabbits in the UTHSCT vivarium [12]. Vivarium personnel involved in the care of rabbits in any capacity, including procedural techniques, anesthesia, or euthanasia, were trained to vivarium and our Institutional Animal Care and Use Committee (IACUC) standards. All agents administered to the rabbits were of pharmaceutical grade, as previously described [12]. In our study, a total of n = 90 rabbits were involved. Of these, n = 41 were treated with the investigational drug, and the remaining n = 49 were designated as donors (Figure 1). Table 1 below provides a breakdown of the rabbits used in each experimental group. Rabbits that were excluded from the experiment due to chest tube crossover and infection are marked with a red asterisk (*).

### 4.2. Non-Survival Blood Collection

These procedures were conducted as we previously reported [12]. After exsanguination, all rabbits were assessed for the absence of spontaneous breathing, heart beats, and corneal reflexes to confirm death. Homologous rabbit blood was immediately mixed with 5% citrate (1:7 ratio) and placed on a rocker to prevent coagulation.

### 4.3. Chest (Thoracostomy) Tube Placement, Retention, and Maintenance

We developed a novel strategy to place and retain a thoracostomy tube in rabbits with RH, as previously reported [12]. In the current project, we used the same techniques to place and secure the chest tube via suturing and vest placement. CT was used to verify chest tube placement. The residual pneumothorax was aspirated via a chest tube to ensure comfortable breathing. Chest tube patency was maintained via a daily saline flush (1.2 mL).

### 4.4. RH Injury Induction and Evaluation

We accomplished these techniques as previously reported [12]. Donor blood was administered in three different schedules (Figure 1, blood drops) via chest tube, delivering a total of 120 mL. Before delivery, the citrated donor blood was recalcified (CaCl_2_·10 mM, American Regent, Inc., Shirley, NY, USA) and supplemented with thrombin (0.3 U/mL) (Sigma-Aldrich, Saint Louis, MO, USA) to promote coagulation. Chest ultrasonography ( Logiq e, s R10 v1.3, GE Healthcare, Chicago, IL, USA) was performed daily to visualize and monitor the pathogenesis of RH as previously reported (Figure 1) [12]. Using the built-in distance measurement tool, the 2D Clot area (cm^2^) was measured before (0 h) and after IPFT (24 h) (Figure 2B). The baseline (0 h) and final (24 h) raw data were recalculated using the percent decrease formula, now represented as clot size reduction (%) (Figure 2B). CT was performed before and after drainage at 24 h to evaluate the extent of lung restriction and expansion. for each treatment group (Figure 1, Day 8). Vivoquant 3.0 software was used to view/edit CT scans and to calculate the total lung volume (mm^3^).

### 4.5. Gross and Histological Evaluation of Retained Hemothorax Injury

Euthanasia procedures were performed as previously reported [12]. To reduce potential observational bias, a surgeon uninvolved in the delivery of interventional agents (A.O.A.) assessed the gross presentation of RH after treatment via IPFT. The following hallmarks were evaluated: clot volume, pleural adhesion and organization, sanguineous pleural effusion, and overall lung morphology. The clots were surgically harvested and placed in a saline bath. The volume displaced after adding the clot was measured and classified as clot volume (mL). The absence of clot in the pleural space was assigned a value = 0 mL. Lung tissues were harvested and prepared for histologic and immunohistochemical analyses as previously performed [12].

### 4.6. IPFT Dosing Regimen

The following agents were delivered into the pleural space of rabbits with RH via chest tube: PBS (negative control), sctPA, and scuPA. scuPA (LTI-01) was obtained from Lung Therapeutics, Inc., while sctPA (Activase, Genentech, San Francisco, CA, USA) is commercially available for intravascular clot lysis and is currently used off-label for intrapleural application. Interventional drugs were mixed with PBS (vehicle) right before delivery into the pleural space. A single bolus injection of either 2 mg/kg scuPA or sctPA was delivered at 0 h. For the double dose schedule, a half dose of either 1 mg/kg sctPA or scuPA was delivered at 0 h and another half dose at 8 h. Tube thoracostomy was attempted at 8 h and 24 h to determine pleural drainage success, evacuate pleural effusions, and alleviate dyspnea. Dose de-escalations were limited to rabbits treated with 0.5 mg/kg scuPA or a total of 1 mg/kg total dose in order to conserve animals and because administration of sctPA was a comparator to our primary focus to determine the ability of scuPA to clear RH.

### 4.7. Cytological and Biochemical Markers in RH PFs

Cytological quantification techniques were employed to measure the RBC and WBC counts and WBC distribution in RH PFs, as we previously reported [12]. Inflammatory cytokines such as IL-6 (R&D Systems, Minneapolis, MN, USA), IL-8 (RayBiotech, Peachtree Corners, GA, USA), TNF-α (R&D Systems, Minneapolis, MN, USA), and TGF-β (R&D Systems, Minneapolis, MN, USA) in the RH PFs were measured by ELISA as we previously reported [12]. Likewise, fibrinolytic components in the RH PFs were also measured, including Active and total antigen PAI-1 (Innovative Research, Pleasanton, CA, USA), active and total antigen tPA (Innovative Research, Pleasanton, CA, USA), and active and total antigen uPA (Innovative Research, Pleasanton, CA, USA). We previously determined that there is no cross-reactivity between human and rabbit uPA or tPA antigens and analyses of changes in the interventional agents in rabbit pleural fluids were thus limited to rabbits receiving human scuPA or sctPA.

### 4.8. Anesthesia

As we previously reported, all personnel involved in delivering anesthesia were fully trained to vivarium and IACUC standards, and the agent used was included. Buprenorphine (Stephenson’s Pharmacy, TX, USA) 0.05 mg/kg IM (0.03–0.1 mg/kg) and Midazolam (Avet Pharmaceuticals Inc., East Brunswick, NJ, USA) 0.5 mg/kg IM (0.05–2.0 mg/kg) [12]. Notably, the dosing schedules we used were based on our several decades of experience in mitigating adverse events. The use of ketamine (Zoetis Animal Health, Parsippany-Troy Hills, NJ, USA), 50 mg/kg (20–60 mg/kg) was limited to donor rabbits. As we previously reported, there were no apparent serious drug interactions [12]. A veterinarian monitored all experiments and the potential need for drug adjustments.

### 4.9. Post-Procedure Monitoring, Nursing, and Support

As we previously reported [12], rabbits were monitored every 6–12 h from the induction of RH until the conclusion of each experiment. During this time, rabbit well-being was assessed using a battery of physiologic assessments, including cardiovascular and systemic determinations [12]. Post-surgical support included pain medications and comfort measures, among other modalities we previously reported [12].

### 4.10. Humane Endpoints

These were monitored continuously, as we previously reported [12]. These assessments included extreme lethargy and indices of cardiovascular dysfunction [12]. If interventions designed to address such abnormalities were ineffective, affected subject animals were euthanized by the veterinarian or approved vivarium staff [12].

### 4.11. Statistics

This study was sufficiently powered to evaluate the outcomes using six rabbits per group based on our prior experience [10]. Kruskal–Wallis (Unpaired, Nonparametric, One-Way ANOVA) or Friedman (Paired, Non-parametric, One-Way Anova) tests were used to determine the statistical significance and to compare the mean ranks of a data set with more than two groups. Moreover, Dunn’s multiple comparison test was used to adjust for the *p*-value. The Mann–Whitney test (Unpaired, Nonparametric, *t*-test) was used to determine the statistical significance of datasets with only two groups. Data were presented as box and whisker plots, while the graphs were generated using GraphPad Prism v 9.3.1.

## 5. Conclusions

This is, to our knowledge, the first interventional study to test the use of IPFT in a new rabbit model of RH. We found that scuPA and sctPA were comparably effective in clearing RH in rabbits. Two daily doses of 1 mg/kg were effective in substantively clearing and organizing intrapleural clot burden. Pleural drainage was increased by the intrapleural administration of either agent. Both were well tolerated. The effective dosing range in rabbits was comparable to that used in empyema in the rabbit, suggesting that a range of similar doses can be tested in patients with florid organization, loculation, and failed drainage in RH.

## Figures and Tables

**Figure 1 ijms-25-08778-f001:**
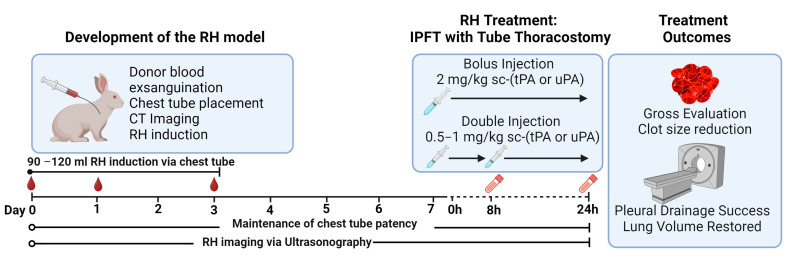
Schematic representation of the RH model development, treatment, and outcome evaluation. This figure was created using https://Biorender.com.

**Figure 2 ijms-25-08778-f002:**
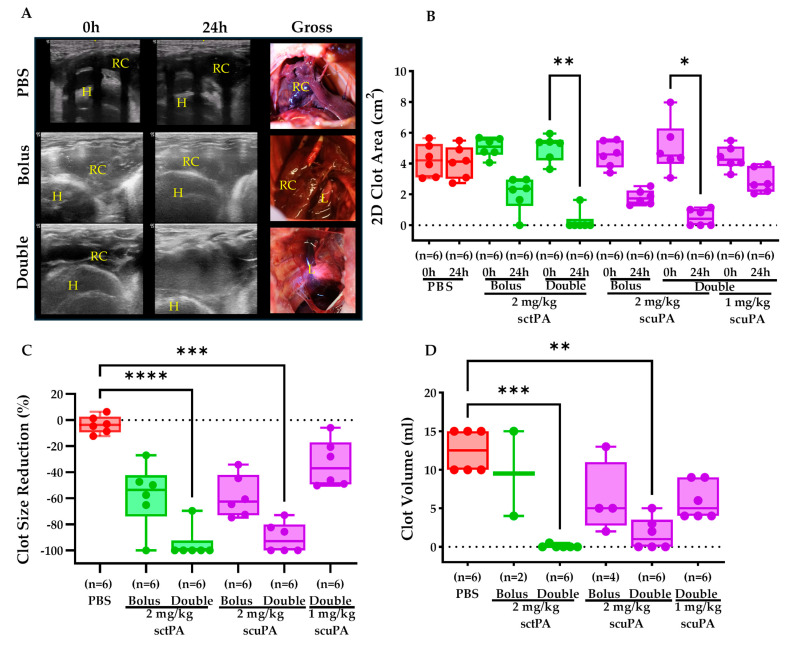
Retained clot size evaluation after treatment with IPFT. (**A**) Representative sonographs demonstrate the presence of clot before (0) and after (24 h) treatment with PBS (negative control), bolus, or double injection of fibrinolytics. Gross images taken postmortem show the presence of retained clot (RC) and lung (L) in the pleural space. (**B**) Based on sonographic imaging and using the distance measurement tool (Logiq e, R10 v1.3, GE Healthcare, Chicago, IL, USA), we measured the 2D Clot area (cm^2^) in the pleural space of rabbits before (0 h) and after treatment (24 h) with either PBS, 2 mg/kg sctPA (Bolus or Double delivery), 2 mg/kg scuPA (Bolus or Double delivery), or 1 mg/kg scuPA (Double delivery). (**C**) Using the same raw data set, the percent decrease in clot size was calculated between 0 h and 24 h. (**D**) The clots were harvested postmortem and measured using volume displacement method as described in material and method section. Clot volume data for 2 mg/kg sctPA (Bolus) and scuPA (Bolus) have limited data points as clots were preserved in situ at the beginning of the project to visualize the clot–lung adherence via histology. Kruskal–Wallis test with Dunn’s multiple comparison test was used to determine statistical significance for all data sets. A *p* < 0.05 was represented as *, *p* < 0.01 as **, *p* < 0.001 as ***, and **** as *p* < 0.0001.

**Figure 3 ijms-25-08778-f003:**
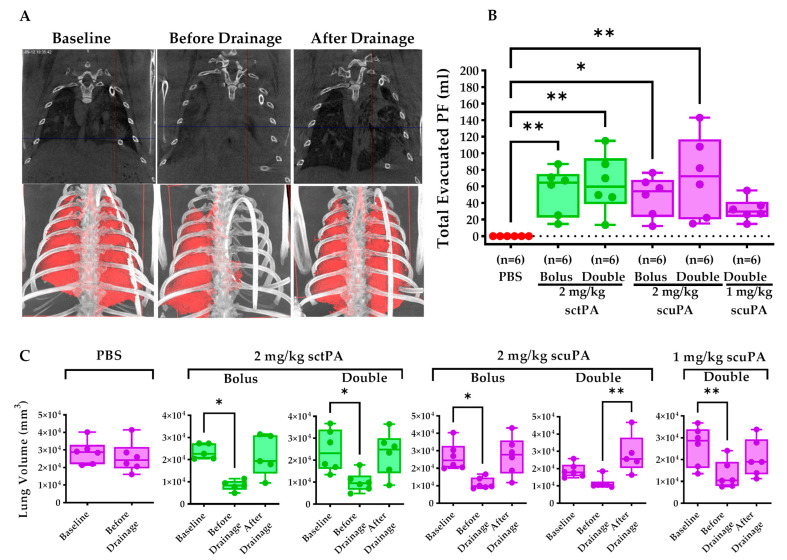
Evaluation of lung restriction and expansion via CT. (**A**) Representative CT images (coronal view) of the rabbit thorax before RH induction (baseline) and before and after drainage at 24 h post-treatment are shown. Rabbits treated with PBS failed to drain. Therefore, these rabbits did not undergo a post-drainage scan. (**B**) Total evacuated PFs were calculated by summing the PF volumes drained at 8 h and 24 h; moreover, failure to drain any PFs from the rabbits were assigned a value = 0. (**C**) Vivoquant Studio 3.0 was used to calculate the 3D rabbit lung volume (mm^3^) for each treatment group at baseline (before induction of RH), before drainage, and after drainage. The Kruskal–Wallis or Friedman test with Dunn’s multiple comparison test was used to determine the statistical significance between treatment groups. A *p* < 0.05 was depicted as * and *p* < 0.01 as **.

**Figure 4 ijms-25-08778-f004:**
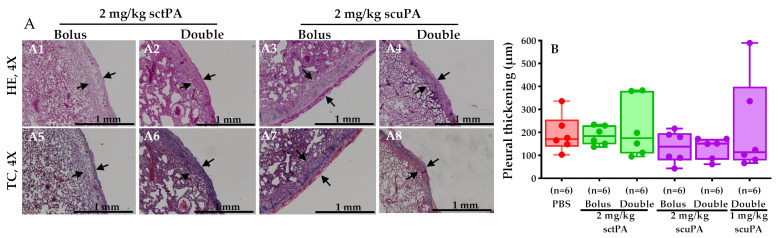
Histological evaluation of the rabbit pleural lining after treatment with IPFT. (**A**) Injured lung tissue from each rabbit subject were stained with Hematoxylin and Eosin Stain (**A1**–**A4**) to visualize pleural thickening (black arrows, 1 mm scale) and Masson Trichrome Stain (**A5**–**A8**) to visualize the overexpression of collagen (blue lining, 1 mm scale) in the pleural lining. (**B**) Morphometry quantified the extent of pleural thickening from each rabbit treated either PBS, sctPA, or scuPA. The Kruskal–Wallis test with Dunn’s multiple comparison determined the statistical significance between treatment groups. A *p* > 0.05 demonstrated no statistical significance (not shown).

**Figure 5 ijms-25-08778-f005:**
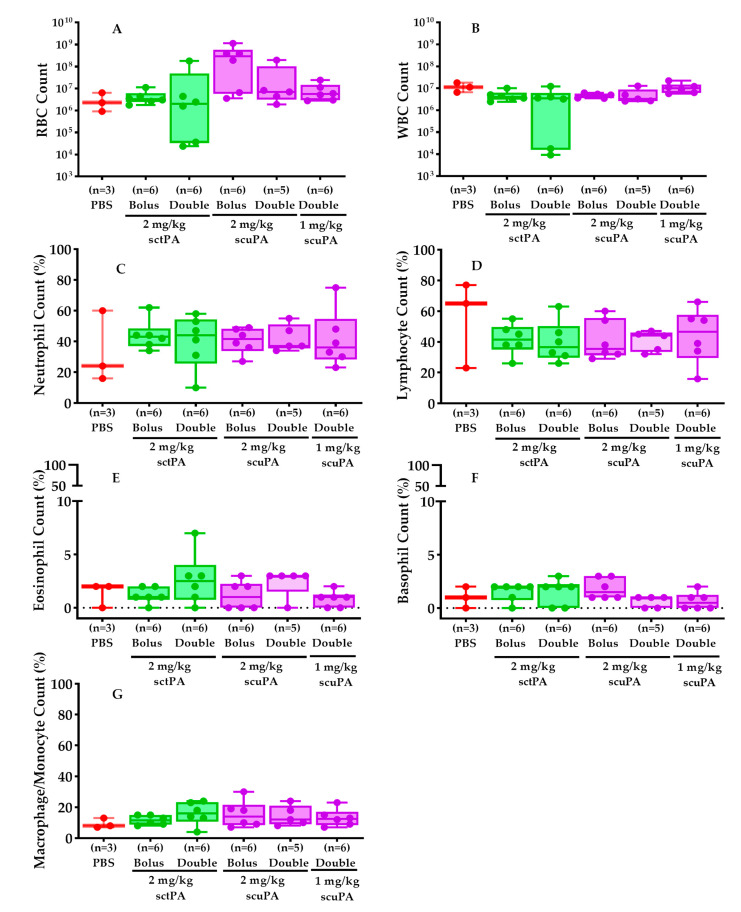
IPFT did not increase extravascular bleeding and WBC infiltration in RH PFs. Total red (RBCs, (**A**)) and white (WBCs, (**B**)) blood cells in the RH PFs were measured to determine whether IPFT increased extravascular bleeding and WBC infiltration. WBC differential staining and analyses (**C**–**G**) was performed on the treated RH PFs as previously performed [12,15]. The Kruskal–Wallis test with Dunn’s multiple comparison determined the statistical significance between treatment groups. A *p* > 0.05 was not shown.

**Figure 6 ijms-25-08778-f006:**
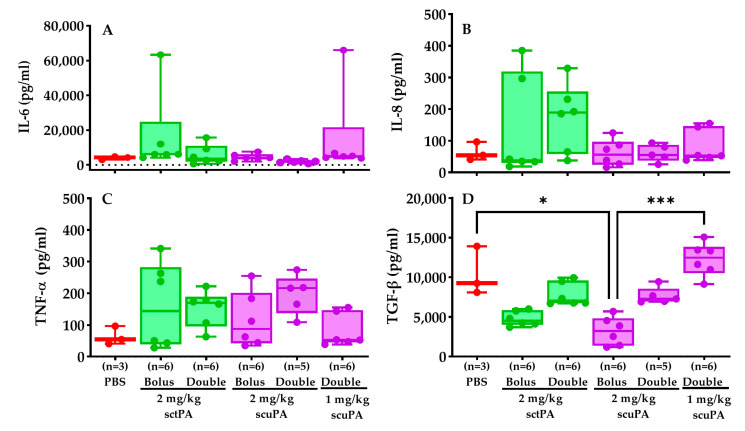
Effects of IPFT on the inflammatory profile of rabbits with RH. IL-6 (**A**), IL-8 (**B**), TNF-α (**C**), and TGF-β (**D**) levels in the treated RH PFs were quantified by ELISA as previously described [12,15]. The Kruskal–Wallis test with Dunn’s multiple comparison determined the statistical significance between treatment groups. A *p* < 0.05 was represented as * and *p* < 0.001 as ***.

**Figure 7 ijms-25-08778-f007:**
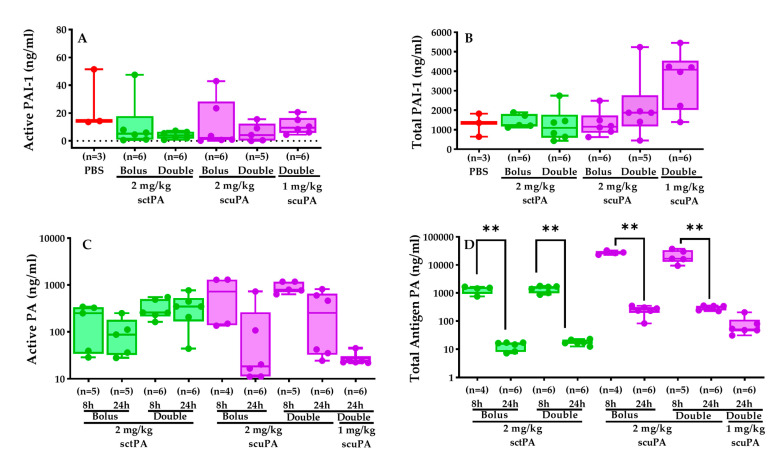
Intrapleural levels of plasminogen activators (tPA and uPA) and PAI-1 during IPFT of RH model. Levels of active (**A**) and total PAI-1 (**B**) in the PFs at 24 h were measured using ELISA as previously described [12,15]. Levels of active (**C**) and total antigen (**D**) PA (human tPA or uPA) in the RH PFs collected at 8 h and 24 h after administration pf the human PAs were measured by ELISA. The Kruskal–Wallis test with Dunn’s multiple comparison determined the statistical significance between treatment groups. A *p* < 0.01 was represented as **.

**Table 1 ijms-25-08778-t001:** Distribution of rabbits used for IPFT.

Investigational Drugs	Number of Rabbits Received Investigational Drugs	Number of Rabbits Used as Donors in the Experiment
PBS	9 − 3 * = 6	11
sctPA Bolus (2 mg/kg)	7 − 1 * = 6	8
sctPA Double (2 mg/kg)	6	7
scuPA Bolus (2 mg/kg)	6	7
scuPA Double (2 mg/kg)	7 − 2 * = 5	9
scuPA Double (1 mg/kg)	6	7

* Shows the number of rabbits excluded from the experiment.

## Data Availability

All data are shared according to the NIH Resource Sharing statements per funded grants.

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
