# Peer review of "Intrapleural Fibrinolytic Interventions for Retained Hemothoraces in Rabbits"

_ijms, 2024, doi:10.3390/ijms25168778_

Round 1

Reviewer 1 Report

Comments and Suggestions for Authors

I had the pleasure of revising the manuscript entitled “Intrapleural Fibrinolytic Interventions for Retained Hemotho- 2 races in Rabbits”. I found it very interesting, well-planned and described. Please, find below  my comments and suggestions

-line 36 – I think it’s reasonable to explain the abbreviation “RH” (it’s explained in the abstract, but should be also, in my opinion, when first used in the introduction section)

-line 42-43 –“…consensus of recent literature  currently supports the initial use of IPFT to clear RH.”  please provide references

-lines 73 and 86 – it’s not clear if one chest drain was inserted to provide blood to rabits’ pleura cavity, then its patency was maintained by saline flush to use it for treatment of RH  or drain was inserted twice. Line 86 is not clear to me.

-line 85-86 -if a drug was given at 0 and 8 h, how long was the drain closed? It’s written that “Chest tube thoracostomy was performed at 8hand 24h to drain and collect RH PFs”

-line 89-90 – “Histology was performed on injured lung tissues to evaluate pleural thickening and collagen deposition.”  - why “injured tissue”?

-if I understand correctly, blood was inserted into the pleural cavity, without any injury to the patient's pleura or chest wall vessels. I ‘m wondering, whether in the case of real hemothorax with vessel injury administration of IPFT could cause an increased risk of extravascular bleeding. Could you speculate on that in your manuscript?

-page 8, section 2.7 -how would the authors explain the difference in concentration of active plasminogen measured post 24 h between sctPA and scuPA (Fig7c), as well as differences regarding total PAI-1 between single and double doses of scuPA and sctPA (Fig 7b). May it have any impact on which substance to use?

-line 481-  “De Vera, 2024}” - I guess the number of reference is missing

I would like to congratulate the authors on their great work and I keep my fingers crossed for quick publication.

Author Response

General Comments: I had the pleasure of revising the manuscript entitled “Intrapleural Fibrinolytic Interventions for Retained Hemotho- 2 races in Rabbits”. I found it very interesting, well-planned and described. Please, find below  my comments and suggestions

Response:  The favorable comments are very much appreciated.

Comment 1: Line 36 - I think it's reasonable to explain the abbreviation "RH" (it's explained in the abstract, but should be also, in my opinion, when first used in the introduction section).

Response 1:  Agreed and done on  Page 1, Paragraph 1, lines 36-37 as requested.

Comment 2: Line 42-43 -" ... consensus of recent literature currently supports the initial use of IPFT to clear RH." please provide references

Response 2: The references have been added as requested, Page 1, Paragraph 1, line 43 of the revised text.

Comment 3: Lines 73 and 86 - it's not clear if one chest drain was inserted to provide blood to rabbits' pleura cavity, then its patency was maintained by saline flush to use it for treatment of RH or drain was inserted twice. Line 86 is not clear to me.

Response 3: We added additional text on lines to clarify that a single chest tube was placed on Page 2 , Paragraph 2, lines 79-81, with additional text added on Page 2, Paragraph 2, line 82, and  Page 2, Paragraph 3, lines 84-85 and 89-91 to clarify how the chest tube was maintained and used for drainage. 

Comment 4: Line 85-86 -if a drug was given at 0 and 8 h, how long was the drain closed? It's written that "Chest tube thoracostomy was performed at 8h and 24h to drain and collect RH PFs".

Response 4: We amended the text on Page 2, Paragraph 3, lines 89-92 to address this point and clarify how drainage was performed.

Comment 5: Lines 89-90 - Histology was performed on injured lung tissues to evaluate pleural thickening and collagen deposition." - why "injured tissue?

Response 5: The term 'injured" has been removed in Page 2, Paragraph 3, Lines 94-95to address this point, as requested.

Comment 6: If I understand correctly, blood was inserted into the pleural cavity, without any injury to the patient's pleura or chest wall vessels. I 'm wondering, whether in the case of real hemothorax with vessel injury administration of IPFT could cause an increased risk of extravascular bleeding. Could you speculate on that in your manuscript?

Response 6: This is an excellent point. While IPFT is generally well tolerated, bleeding remains a potential complication and clinical concern. We speculate about the issue as to how IPFT is tolerated clinically and particularly where more trauma could occur in the revised Discussion on Page 11, Paragraph 2, lines 280-282.

Comment 7: Page 8, section 2.7 -how would the authors explain the difference in concentration of active plasminogen measured post 24 h between sctPA and scuPA (Fig7c), as well as differences regarding total PAl-1 between single and double doses of scuPA and sctPA (Fig 7b). May it have any impact on which substance to use?

Response 7: We appreciate the reviewer for raising this incisive point. Levels of PA rather than plasminogen are illustrated in Fig 7, and we modified the text of the legend to fig. 7 to clarify this point. The trends, which lack significance, in scuPA activity (Fig7C) were not seen with delivery of sctPA. While the precise underlying mechanisms remain unclear, the decrements in scuPA activity and significant decrements in total scuPA and sctPA may relate to formation of PA-PAI-1 complexes and/or their clearance, now added to Discussion Page 12, Paragraph 1, lines 340-343. 

Comment 8: Line 481- "De Vera, 2024}" - I guess the number of (the) reference is missing.

Response 8: We respectfully offer that the reference is provided with the appropriate citation on Page 15, Paragraph 2, Line 481.

Reviewer 2 Report

Comments and Suggestions for Authors

De Vera et al. tested the ability of intrapleural administration of fibrinolysins sctPA and scuPA to clear RH, with regard to effective dosage range, safety and schedules using a rabbit RH model. They showed that sctPA and scuPA effectively accelerated clearance of intrapleural collections of organizing clots and improves lung restriction. The data were well organized, clearly presented and the manuscript was very well written.

Comments:

1. Fig7 C &D, no data was shown for the PBS group.

2. The lung volumes after drainage in the fibrinolysins groups seemed comparable to that of lung volume in the baseline PBS group. However, if fibrinolysins successfully achieved clot lysis (fig2d) and allowed more thorough drainage of PF(fig3b), wouldn’t this lead to less occupation of intrapleural space by PF and subsequently larger lung volume after drainage?

3. what is the reason and possible impact of the lower TGF-beta level with lower single dose of fibrinolysins, please briefly discuss.

4. page11, line306-308, data of histological analyses was not shown anywhere within the manuscript.

Author Response

General Comment: De Vera et al. tested the ability of intrapleural administration of fibrinolysins sctPA and scuPA to clear RH, with regard to effective dosage range, safety and schedules using a rabbit RH model. They showed that sctPA and scuPA effectively accelerated clearance of intrapleural collections of organizing clots and improves lung restriction. The data were well organized, clearly presented and the manuscript was very well written.

Response: The favorable overall assessments are appreciated.

Comment 1: Fig7 C &D, no data was shown for the PBS group.

Response 1: We did not perform the PA assays, which are directed to human tPA or scuPA in the PBS animals since they did not receive human plasminogen activators.  We previously determined that there is no cross reactivity between rabbit uPA by or tPA and the human PA proteins and thus, we did not perform PA analysis in control PBS treated animals, which did not receive any human PAs. The responses of the human PAs within the rabbit pleural fluids to the one or two (double) human PA formats of administration are supplied in the figure as discussed in the legend. The legend to Fig 7 has been modified now on  Page 10, Paragraph 2, lines 258 and 259 to more clearly address this point. The lack of cross reactivity between human and rabbit PAs has been added in the Methods section on Page 15, Paragraph 2, lines 484-487.

Comment 2: The lung volumes after drainage in the fibrinolysins groups seemed comparable to that of lung volume in the baseline PBS group. However, if fibrinolysins successfully achieved clot lysis (fig2d) and allowed more thorough drainage of PF (fig3b), wouldn't this lead to less occupation of intrapleural space by PF and subsequently larger lung volume after drainage?

Response 2: The reviewer’s inference would be correct if the comparator was the PBS group, which we respectfully offer was not the case. Each animal had a determination of lung volume prior to induction of RH as stated in the original legend to Fig 3. The data in this figure show that in the aggregate, lung volume was restored to levels approximating those prior to induction of RH. To address this point further, the text of the legend to Fig 3 has been modified for better clarity on line Page 6, Paragraph 1, lines 170-171.

Comment 3: What is the reason and possible impact of the lower TGF-beta level with lower single dose of fibrinolysins, please briefly discuss.

Response 3We now address the potential impact of the reduced levels of TGF-beta as requested on Page 12, Paragraph 2, lines 332-336 of the revised Discussion. Levels of TGF-b were relatively lower in rabbits treated with 2 mg/kg single bolus sctPA or scuPA. While the cause of the decrements remains unclear, lower levels of this growth factor could conceivably influence pleural organization, mesomesenchymal transition or neovascularization although RH dissolution was observed in rabbits treated in this manner.

Comment 4: Page 11, line 306-308, data of histological analyses was not shown anywhere within the manuscript. 

Response 4: We thank the reviewer for pointing out this omission and now indicate that the data is not shown now on Page 12, Paragraph 1, line 318 of the revised Discussion as requested.